# Association of Antibody Responses to *Helicobacter pylori* Proteins with Colorectal Adenoma and Colorectal Cancer

**DOI:** 10.3390/pathogens13100897

**Published:** 2024-10-14

**Authors:** Flavia Genua, Julia Butt, Harsha Ganesan, Tim Waterboer, David J. Hughes

**Affiliations:** 1Molecular Epidemiology of Cancer Group, School of Biomolecular and Biomedical Science, UCD Conway Institute, University College Dublin, D04 V1W8 Dublin, Ireland; flaviagenua@rcsi.ie (F.G.); harsha.ganesan@ucd.ie (H.G.); 2School of Pharmacy and Biomolecular Sciences, RCSI University of Medicine and Health Sciences, D02 VN51 Dublin, Ireland; 3Infections and Cancer Epidemiology, German Cancer Research Center (Deutsches Krebsforschungszentrum, DKFZ), 69120 Heidelberg, Germany; j.butt@dkfz-heidelberg.de (J.B.); t.waterboer@dkfz-heidelberg.de (T.W.)

**Keywords:** serology, colorectal cancer, colorectal neoplasms, *Helicobacter pylori*

## Abstract

*Helicobacter pylori* (*H. pylori*) has been implicated in colorectal carcinogenesis. Here, the association of immune responses to bacterial exposure with advancing stages of colorectal neoplasia was assessed by multiplex serology. Immunoglobulin (Ig) A and G antibody responses to thirteen proteins of *H. pylori* were measured by a Luminex-based multiplex assay in plasma from patients with colorectal cancer (CRC, n = 25), advanced adenoma (n = 82), or small polyps (n = 85) and controls (n = 100). Multivariable logistic regression was used to assess the association of bacterial seropositivity with colorectal neoplasia. The threshold for overall seropositivity required subjects to be positive for at least 4 out of the 13 tested antigens. In a cohort subset with matched data (n = 34), *H. pylori* seropositivity was correlated with bacterial abundance in both neoplastic and matched normal tissue. While no association was found between *H. pylori* seropositivity and the presence of CRC, IgA seropositivity to CagA was associated with a decreased risk of advanced adenoma (odds ratio, OR = 0.48, 95% confidence intervals, CIs: 0.24–0.96). Regarding IgG, higher antibody responses to HpaA was associated with advanced adenoma occurrence (OR = 2.46, 95% CI: 1.00–6.01), while responses to HP0395, CagA and Catalase were associated with polyp development (OR = 2.65, 95%, CI: 1.31–5.36, OR = 1.83, 95% CI: 1.01–3.32, and OR = 2.16, CI: 1.09–4.29, respectively). Positive correlations were found between *H. pylori* abundance in the normal mucosa and levels of both the IgA and IgG antibody response to Catalase and VacA antigens (r = 0.48, *p* < 0.01; r = 0.37, *p* = 0.04; r = 0.51, *p* < 0.01; and r = 0.71, *p* = 0.04, respectively). Conversely, *H. pylori* abundance was negatively correlated with levels of IgA antibody response to HpaA and with IgG antibody response to HP0231 in the diseased tissue (r = −0.34, *p* = 0.04 and r = −0.41, *p* = 0.01, respectively). The association between levels of *H. pylori* antigens and colorectal neoplasia risk gradually decreased with the adenoma progression, implicating the early activation of the immune response at the polyp stage. Thus, the evaluation of antibody response to certain bacterial antigens may indicate the presence of early-stage colorectal neoplasia. Further studies are needed to clarify the role *H. pylori* or the immune response to its antigens may have in colorectal carcinogenesis stages.

## 1. Introduction

Colorectal cancer (CRC) is the second leading cause of cancer-related fatalities globally and ranks third in terms of prevalence, with 1.8 million new cases recorded in 2018 [1]. Growing evidence underscores the multifaceted nature of CRC aetiology, implicating genetic predisposition, environmental factors, metabolic irregularities, microbiome composition, and compromised gut barrier integrity [2,3]. Additionally, persistent inflammation resulting from infections is believed to fuel the progression of CRC [4]. Our group previously showed that that seropositivity to certain *Streptococcus gallolyticus subsp. gallolyticus* (SGG) and *Fusobacterium nucleatum* (*F. nucleatum*) proteins was associated with the presence of advanced stages of colorectal neoplasia, including colorectal adenoma (CRA) and CRC [5]. Thus, the evaluation of antibody response to bacteria may provide an insight into the temporal relevance of microbial exposures in colorectal carcinogenesis and serve to identify individuals at increased risk for developing CRC or to detect the presence of CRC in the early stages.

*H. pylori* is the major aetiological cause of gastric cancer, likely via the promotion of chronic inflammation through an increase in gastrin secretion or toxin production. Although *H. pylori* infection is mostly observed in the stomach, increasing epidemiological evidence also suggests a link with various extragastric diseases, including CRC [6]. This association was demonstrated by an in vivo study showing that *H. pylori* infection accelerated tumour growth in adenomatous polyposis coli mouse models (Apc^+/−^ and Apc^+/1638N^) [7].

Although meta-analyses support an increased risk of CRA and CRC associated with *H. pylori* infection [8,9,10], the differences in study methodology and the lack of control for confounders are important caveats precluding convincing conclusions on the aetiological involvement of this bacterium in colorectal carcinogenesis. A few prospective studies assessed the risk of CRC development with *H. pylori* seropositivity, but the findings were inconsistent and some of them reported null associations [11,12]. Using multiplex serology, two studies from independent populations in the United States found that higher antibody levels to HcpC and VacA from *H. pylori* showed a strong association with CRC [13,14], and similar results were found in a nested case–control study conducted in a large European cohort [15]. Interestingly, the association between *H. pylori* and CRC also varied, with results depending on the race or ethnicity of the population in which the studies were conducted [11,16].

Here, due to limited knowledge of the immune response to these bacterial antigens in colorectal neoplastic progression, we assessed whether antibody responses to *H. pylori* proteins are associated with stages of neoplasia development from small polyps to more advanced adenomas and cancers in a patient case–control study conducted within a CRC population.

## 2. Materials and Methods

### 2.1. Clinical Characteristics

This study included 292 individuals from Ireland who donated blood samples prior to bowel preparation for colonoscopy following a positive result for an immunochemical faecal occult blood test (FIT) or prior to the surgical resection of a colorectal tumour. Patients with CRC (n = 25), advanced adenoma (n = 82), and polyps (n = 85) were diagnosed at the Departments of Gastroenterology and Surgery, Tallaght University Hospital (TUH), in Dublin, Ireland (most of the patients with CRC were recruited through the surgery department). Controls (n = 100) were individuals with no colorectal neoplasia detected upon colonoscopy (‘colonoscopy-negative’ controls). All CRCs were classified according to the tenth revision of the International Classification of Diseases (ICD-10). Advanced adenoma includes adenomas with high-grade dysplasia (HGD), adenomas with at least 20% tubular villous or villous features, all adenomas greater than 10 mm, and the presence of three or more adenomas [17,18]. The clinical data, including age at diagnosis, sex, pTNM (tumour stage, regional lymph node involvement, and distant metastasis) staging, and primary tumour localization were taken from patient medical records (see Table 1 for a summary of the clinical characteristics of our study cohorts). All patients gave informed consent in accordance with the Helsinki Declaration, and all patient samples were pseudonymized to protect participant identity. The study was approved by the Ethical Committee of the St. James’s Hospital and Federated Dublin Voluntary Hospitals Joint Research Ethics Committee (Ireland, reference 2007-37-17).

### 2.2. Sample Collection

The blood samples were collected within one day prior to surgery or colonoscopy in 6 mL VACUTAINER^®^ tubes (Cruinn Diagnostics, Dublin, Ireland) with EDTA. Within 4 h of collection, bloods were centrifuged at 2000× *g* for 10 min to separate the top plasma layer, which was then stored at −80 °C in cryovials. Disease and matched normal mucosal tissue samples were collected during a resection of the primary tumour or by biopsy, before treatment, while all adenoma biopsies were obtained at colonoscopy during a pilot CRC screening programme as described previously [19].

### 2.3. Multiplex Serology

Plasma samples (~20 uL) were analysed for antibody responses against 13 proteins from *H. pylori* using multiplex serology performed in a fluorescent bead-based suspension array, as described previously [20]. Briefly, antigens were expressed as Glutathione-S-transferase (GST)-tagged fusion proteins and affinity-purified on glutathione-casein coupled polystyrene beads (Luminex Corp, Austin, TX, USA) with distinct internal fluorescence [20]. After the pre-incubation step, the plasma samples were incubated with the antigen-loaded bead mixture and bound IgG or IgA antibodies were labelled separately by biotinylated secondary antibodies (goat anti-Human IgG-Biotin #109-065-098 and goat anti-Human IgA-Biotin #109-065-011, Jackson ImmunoResearch, Westgrove, PA, USA) and a subsequent incubation with Streptavidin-R-Phycoerythrin (MossBio, Pasadena, MD, USA). A Luminex 200 Analyzer (Luminex Corp., Austin, TX, USA) was then used to distinguish the bead sets and their respective antigens and to quantify the amount of plasma IgG or IgA bound to the antigen. The level of antibody response was given as the median fluorescence intensity (MFI) of at least 100 beads per type measured. Background values against the GST-tag, as well as the bead surface and secondary reagents were subtracted to generate net MFI values.

Antigen-specific cut-offs were defined at the approximate inflexion point of frequency distribution curves under the assumption that a sudden rise in the distribution of antibody response over the percentile of plasma indicated a cut-off for seropositivity, as described previously [21]. Antigen-specific cut-offs with putative protein function are listed in Appendix A. Overall seropositivity to *H. pylori* was defined as those subjects seropositive for at least 4 out of the 13 *H. pylori* proteins included in the multiplex serology panel, as these had previously indicated the best specificity and sensitivity when the assay was validated against a commercially available ELISA [22].

### 2.4. DNA Extraction from Colorectal Tissue Biopsies and Quantitative Real-Time Polymerase Chain (qPCR)

To address whether the observed antibody responses to *H. pylori* reflect its presence in the colorectal tract rather than from other potential infection sites, we also correlated the immune responses to the bacterium with existing matched data for 34 subjects on the relative abundance of *H. pylori* in colorectal neoplasia fresh-frozen tissue and in the respective normal adjacent mucosa. For the DNA extraction, 20–30 mg of tissue was lysed on ice in 400 μL of lysis buffer (50-mmol/L HEPES pH 7.5, 150-mmol/ L NaCl, 5-mmol/L EDTA) and protease inhibitor (Calbiochem, Hampshire, UK), followed by sonication on ice for 3 × 30 s. Lysates were centrifuged at 10,000× *g* for 10 min at 4 °C. DNA was then extracted using the Norgen Biotek All-in-One Purification Kit (Thorold, ON, Canada)). DNA was quantified using a NanoDrop 2000 c spectrophotometer (Thermo Scientific, Asheville, NC, USA). DNA extractions were stored at −80 °C. Quantitative real-time polymerase chain reaction (qPCR) to quantify the relative abundance of *H. pylori* in both disease and matched normal tissue from patients with CRA or CRC was performed on the Applied Biosystems 7500 Real-Time PCR System (Thermo Fisher Scientific, Dublin, Ireland). We amplified bacterial DNA using the PowerUp™ SYBR™ Green Master Mix (Thermo Fisher Scientific, Waltham, MA, USA). Each 20 μL reaction consisted of 30 ng of template DNA, 400 nM of each primer set and 10 μL of SYBR Green Master Mix (cat. no. A25742). Samples that showed no amplification within 50 cycles were censored and we assumed that no template was present or below the detection limit. All samples and controls were run in duplicate. Primers were synthetized by Merck Life Science Limited (Vale Road, Arklow, Co. Wicklow, Ireland). To amplify the *CagA* region, previously published primers were used as Forward: 5′-GTTGATAACGCTGTCGCTTC-3′ and Reverse: 5′- GGGTTGTATGATATTTTCCATAA-3′ [23]. The reaction conditions were 94 °C for 3 min, 35 cycles of 94 °C for 1 min, 55 °C for 1 min, 72 °C for 1 min, and 72 °C for 10 min. Products from qPCR (10 ng/μL in a final volume of 15 μL) were sent to Eurofins Genomics (Eurofins Genomics UK Limited, Wolverhampton, UK) for Sanger sequencing as a confirmatory assay. Sequencing data were verified for quality, corrected when possible, and then aligned to the reference genome of the bacterial target using the online BLAST tool [24]. The relative quantification (RQ) of *H. pylori* is calculated by 2^−ΔCT^, where ΔCT is the difference in the copy number threshold (CT) for the test gene (*CagA*) and reference gene (human prostaglandin transporter, *PGT*).

### 2.5. Statistical Analysis

We estimated the association of colorectal neoplasia with respective IgA and IgG seropositivity to individual *H. pylori* proteins using conditional logistic regression models to calculate odds ratios (ORs) and 95% confidence intervals (95% CIs). To address whether minor inflammatory-related conditions could act as confounders for the observed associations, we conducted a sensitivity analysis restricting the control group to those subjects with “no abnormalities detected after colonoscopy” (NAD, n = 37), including haemorrhoids, mild colitis and diverticulosis, and other minor inflammatory conditions. The analyses are adjusted by age and sex and are presented in the text, except where noted, and in the main data tables. The results of the unadjusted analysis are included in the Appendix A (Appendix A). The point biserial test was used to evaluate the correlation between *H. pylori* abundance in both colorectal neoplastic and matched normal tissue and antibody response to the bacterium in plasma (in a smaller cohort of patients with available matched data, n = 34). A multiple-testing adjustment was conducted using the false discovery rate (FDR). Given that the *p*-values are derived from a clear hypothesis-driven approach with a small number of comparisons, we base our interpretation on the observed *p*-values, but to be cautious, we also present the q-values for the multiple-testing correction. *p*- and q-values ≤ 0.05 were considered statistically significant. All statistical analyses were performed with IBM SPSS Statistic for Windows, version 27.0 (SPSS Inc., Chicago, IL, USA) and Rstudio, version 4.0.0 (RStudio Team (2020). RStudio: Integrated Development for R. RStudio, PBC, Boston, MA, USA, URL http://www.rstudio.com/).

## 3. Results

There was no statistically significant association observed between the overall *H. pylori* IgA or IgG seropositivity (based on the stringent positivity cut-off of at least four antigens) with the presence of CRC (Table 2 and Table 3). However, IgG seropositivity was associated with advanced adenoma development (OR = 1.88, CI: 1.03–3.47, Table 3). Additionally, considering single-antigen responses, the IgA reaction to the CagA virulence antigen was associated with a decreased risk of advanced adenoma (OR = 0.48, 95% CI: 0.24–0.96, Table 2). Higher levels of IgG antibody responses to HP0305, CagA and Catalase were associated with polyp development (OR = 2.65, 95% CI: 1.31–5.36, OR = 1.83, 95% CI: 1.01–3.32, and OR = 2.16, CI: 1.09–4.29, respectively, Table 3), while the IgG response to HpaA was associated with advanced adenoma occurrence (OR = 2.46, 95% CI: 1.00–6.01, Table 3).

### 3.1. Sensitivity Analysis Based on the Control Group

To address whether minor inflammatory-related conditions could act as confounders in the association analyses, we conducted a sensitivity analysis restricting the control group to those subjects with “no abnormalities detected after colonoscopy” (NAD, n = 37), including haemorrhoids, mild colitis and diverticulosis, and other minor inflammatory conditions. However, when the analysis was restricted to the NAD group, most of the observed associations did not retain statistical significance but were all in the same risk direction (Appendix A). Additionally, new statistically significant inverse and positive associations were observed with polyp development for the IgA response to HP1564 (OR = 0.09, CI: 0.009–0.98) and the IgG response to GroEL (OR = 2.23, CI: 0.97–5.12), respectively (Appendix A). Although these associations were non-significant when using the full control group, they were in the same respective risk directions (Table 2).

### 3.2. Correlation between H. pylori Tissue Levels and the Antibody Response in Plasma

The relative abundance of *H. pylori* in CRA and CRC disease tissue and the respective normal adjacent mucosa, as previously ascertained by qPCR for 34 patients, showed some significant correlations with the immune responses. Positive correlations were found between *H. pylori* abundance in the normal mucosa with both the IgA and IgG antibody responses to Catalase and VacA antigens (r = 0.48, *p* < 0.01; r = 0.37, *p* = 0.04; r = 0.51, *p* < 0.01; and r = 0.37, *p* = 0.04, respectively), as well as the IgA only antibody response to HyuA (r = 0.36, *p* = 0.05), (Appendix A). Conversely, *H. pylori* abundance was negatively correlated with the IgA and IgG antibody response to HpaA or HP0231, respectively, in the diseased tissue (r = −0.34, *p* = 0.04 and r = −0.41, *p* = 0.01, Appendix A). Correlations between overall seropositivity and *H. pylori* tissue levels were not computed due to the small number of colorectal neoplasia samples (IgA n = 7; IgG n = 12).

## 4. Discussion

We measured antibody responses to *H. pylori* by a separate detection of IgA and IgG. We observed that seroprevalence to some of the bacterial antigens varied significantly between cases with colorectal neoplasia and control groups.

*H. pylori’s* natural habitat is the stomach. However, it is possible that *H. pylori* or its metabolic products could increase the likelihood of developing polyps as the first step towards CRC development, infecting healthy colon tissue prior to the carcinogenic process. *H. pylori* may induce mucosal and systemic antibody responses and potentially cause pro-carcinogenic effects, as previously hypothesised for genotoxic *Escherichia coli* species and enterotoxigenic *Bacteroides fragilis* (ETBF) [21]. It is known that certain strains of *H. pylori* can release a toxin—CagA—that has a direct carcinogenic effect on the gastrointestinal mucosa, resulting in a peptic ulcer, premalignant lesions, and gastric adenocarcinoma [25,26]. Furthermore, it is hypothesised that *H. pylori* can mediate the shaping of the intestinal microbiota, contributing to the increased CRC risk of infected hosts [27], and that the microbe can induce CRC by promoting inflammation through STAT3 signalling and through the loss of goblet cells [7].

We previously showed that IgG and IgA seropositivity to *F. nucleatum* proteins was associated with CRC and that, conversely, seropositivity to SGG was associated with an increased risk for precancerous lesions, leading to the hypothesis that the first bacterium might act as a “passenger bacterium” increasing in abundance due to favourable growth conditions with dysplastic progression, while SGG may be a potential aetiological factor in the transition of a polyp to malignant disease, and its detection could help to identify precursors that may more likely progress to cancer [5]. In this study, overall seropositivity to *H. pylori* proteins were not associated with CRC risk. In agreement with this result, null associations between *H. pylori* and CRC risk were also found in a serological case–control study conducted in Spain [28]. Conversely, a prospective serological study conducted in different populations in the United States found that VacA seropositivity was associated with an increased risk of CRC, especially in African Americans, and a strong dose–response relationship between VacA antibody levels and risk of CRC development was found among all the races/ethnicities combined and among African Americans alone [14]. As hypothesised by the authors, different serological responses to *H. pylori* might be dependent on race/ethnicity, and they did not find any significant associations between *H. pylori* proteins and CRC risk in white Americans, although they observed the same risk direction. Similarly, a prospective European study (EPIC) found that higher antibody levels to HcpC and VacA showed the strongest associations with CRC development [15]. All these studies, using the same methodology employed here, had no information on whether any of the subjects had colorectal neoplasia at the time of blood draw enrolment.

Seropositivity for the IgG response to >3 antigens was associated with an increased risk for the presence of advanced adenomas. Furthermore, the IgG response to HpaA was also positively associated with advanced adenoma development, while responses to HP0305, CagA and Catalase were associated with a higher risk for polyps. Overall, the association between responses to *H. pylori* antigens and colorectal neoplasia risk gradually decreased with the adenoma progression (from polyps to advanced adenoma). One of the hypotheses that might explain this finding is via the immune system recognition of *H. pylori* antigens at the initial stages of carcinogenesis. The early activation of the immune response at the polyp stage might explain the apparent association with a decreased risk of advanced adenoma development as it may parallel the immune-mediated eradication of the infection. Furthermore, this is likely reflected in the more localised mucosal IgA response, as the first line of defence in the resistance against the infection of epithelial cells, compared to the systemic IgG response measurements. The association between *H. pylori* and precancerous lesions has also been evaluated elsewhere. An Iranian study measuring the IgA and IgG antibody responses to *H. pylori* in blood using ELISA found that patients with polyps or CRC showed higher levels of immunoglobulins compared to the control group [29]. In a retrospective study conducted in patients with no history of CRC or CRA who underwent an esophagogastroduodenoscopy and a colonoscopy during a screening examination, *H. pylori* infection was detected using a urease test, and during follow-up, it was observed that patients with a persistent infection had a higher risk of developing CRA than individuals with successful *H. pylori* eradication [30]. A retrospective study conducted in the US among veterans who completed testing for *H. pylori* between 1998 and 2018 found that being positive for *H. pylori* infection was associated with a small but significantly higher CRC incidence and mortality [31]. These findings support the hypothesis that *H. pylori* infection may be associated with colorectal carcinogenesis.

It is unknown whether the observed antibody responses result from other infection sites than the colon or, although less likely, from a cross-reactive antibody response due to infection with other closely related bacteria. To partially address this, we correlated the relative quantification of *H. pylori*, ascertained by the qPCR of their DNA in disease (n = 34) and in matched normal mucosa (n = 29) tissues, with levels of the IgA and IgG response to *H. pylori.* We observed that levels of both the IgA and IgG antibody responses to Catalase and VacA antigens and the IgA response to HyuA were positively correlated with *H. pylori* abundance in the matched normal mucosa tissue of patients with colorectal neoplasia. These results, even if obtained in a modest cohort of samples, suggest that the immune response is not caused by cross-reactivity or infections from other bacteria (at least for these antigens) but may be activated by the presence of *H. pylori* in the normal colorectal mucosa. Conversely, *H. pylori* abundance was negatively correlated with levels of the IgA antibody response to HpaA and with the IgG antibody response to HP0231 in the diseased tissue, indicating that bacterial abundance might decrease because of the activation of the immune response in the neoplastic tissue.

## 5. Conclusions

There were limited associations observed between the overall seropositivity to *H. pylori* and colorectal neoplasia based on the stringent >3-positive-antigens criteria. However, together with the specific antigen associations, the results suggest that the association between seropositivity to *H. pylori* antigens and colorectal neoplasia risk gradually decreased with the adenoma progression. Thus, the evaluation of antibody response to certain bacterial antigens may be a useful resource to identify individuals at an increased risk for developing CRC or to detect the presence of colorectal neoplasia in the early stages. The results may also shed light on interactions between invasive bacterial species, the immune system, and neoplastic development along the adenoma-to-cancer pathway. These findings need to be validated in other settings with increased samples sizes to assess *H. pylori* and other bacteria serology as a potential biomarker for advancing stages of colorectal neoplasia or of the immune response to developing neoplasia.

## Figures and Tables

**Table 1 pathogens-13-00897-t001:** Clinical characteristics of the studied cohort of patients.

				Advanced Adenoma (n = 82)		
		Controls (n = 100)	Polyp ^1^ (n = 85)	Adenoma ^2^ (n = 60)	HGD (n = 22)	CRC (n = 25)
Sex	Female n (%)	53 (53%)	34 (40%)	28 (47%)	8 (36%)	13 (52%)
	Male n (%)	47 (47%)	51 (60%)	32 (53%)	14 (64%)	12 (48%)
Age	Mean (range)	61 (42–75)	62 (44–75)	64 (50–109)	62 (44–84)	66 (36–89)
Localization	Colon/rectum/na	na	58/25/2	38/20/2	13/9/0	20/4/1
Staging	T staging n (T0/T1/T2/T3/T4/Tx/na)	na	na	na	na	1/2/3/11/4/1/3
	N staging n (N0/N1/N2/Nx/na)	na	na	na	na	16/2/3/1/3
	M staging n (M0/M1/Mx/na)	na	na	na	na	6/3/14/2

HGD: high-grade dysplasia; CRC: colorectal cancer; na: not applicable. ^1^ Polyps were generally hyperplastic and less than 2 mm. ^2^ Advanced adenomas include adenomas with high-grade dysplasia (HGD), adenomas with at least 20% tubular villous or villous features, all adenomas greater than 10 mm, and the presence of three or more adenomas.

**Table 2 pathogens-13-00897-t002:** IgA seropositivity to individual *H. pylori* proteins and association with development of polyp, advanced adenoma, and CRC.

Secondary Ab	Antigen		Control (n = 100) n (%)	Polyp (n = 85) n (%)	OR	95% CI	*p*-Value	q-Value	AA (n = 82) n (%)	OR	95% CI	*p*-Value	q-Value	CRC (n = 25) n (%)	OR	95% CI	*p*-Value	q-Value
**IgA**	>3 proteins	−	81 (81)	71 (84)					69 (84)					20 (80)				
		+	19 (19)	14 (16)	0.70	0.31–1.52	0.37	0.37	13 (16)	0.75	0.34–1.64	0.48	0.48	5 (20)	1.00	0.30–2.94	1.00	1.00
	GroEl	−	83 (83)	73 (86)					69 (84)					19 (76)				
		+	17 (17)	12 (14)	0.71	0.31–1.64	0.42	0.74	13 (16)	0.83	0.37–1.88	0.66	0.82	6 (24)	1.26	0.42–3.82	0.67	0.77
	UreA	−	95 (95)	78 (92)					76 (93)					23 (92)				
		+	5 (5)	7 (8)	1.32	0.42–4.13	0.63	0.85	6 (7)	0.95	0.27–3.30	0.94	0.94	2 (8)	1.43	0.30–6.77	0.65	0.77
	HP0231	−	78 (78)	75 (88)					77 (94)					19 (76)				
		+	12 (12)	10 (12)	0.93	0.38–2.23	0.87	0.93	5 (6)	0.39	0.13–1.17	0.09	0.37	6 (24)	1.79	0.58–5.53	0.30	0.77
	NapA	−	95 (95)	83 (98)					73 (89)					23 (92)				
		+	5 (5)	2 (2)	0.45	0.08–2.44	0.35	0.75	9 (11)	2.51	0.79–7.95	0.11	0.37	2 (8)	1.51	0.25–8.93	0.64	0.77
	HP0305	−	97 (97)	85 (100)					81 (99)					25 (100)				
		+	3 (3)	0 (0)					1 (1)	0.32	0.03–3.20	0.33	0.64	0 (0)				
	HpaA	−	89 (89)	79 (93)					76 (93)					24 (96)				
		+	11 (11)	6 (7)	0.54	0.19–1.57	0.26	0.66	6 (7)	0.57	0.20–1.66	0.30	0.64	1 (4)	0.30	0.03–2.49	0.26	0.77
	CagA	−	66 (66)	58 (68)					65 (79)					17 (68)				
		+	34 (34)	27 (32)	0.91	0.48–1.71	0.77	0.89	17 (21)	**0.48**	**0.24–0.96**	**0.03**	0.26	8 (32)	0.88	0.33–2.31	0.88	0.94
	HyuA	−	87 (87)	76 (89)					72 (88)					22 (88)				
		+	13 (13)	9 (11)	0.72	0.29–1.81	0.49	0.74	10 (12)	0.93	0.38–2.28	0.87	0.93	3 (12)	1.36	0.38–4.80	0.62	0.77
	Catalase	−	92 (92)	76 (89)					75 (91)					23 (92)				
		+	8 (8)	9 (11)	0.71	0.28–1.80	0.47	0.74	7 (9)	0.63	0.23–1.69	0.36	0.64	2 (8)	0.61	0.12–3.11	0.56	0.77
	VacA	−	81 (81)	75 (88)					73 (89)					22 (88)				
		+	19 (19)	10 (12)	0.56	0.24–1.30	0.18	0.57	9 (11)	0.53	0.22–1.28	0.16	0.43	3 (12)	0.58	0.15–2.21	0.43	0.77
	HcpC	−	91 (91)	75 (88)					77 (94)					24 (96)				
		+	9 (9)	10 (12)	1.21	0.46–3.22	0.68	0.86	5 (6)	0.67	0.21–2.15	0.51	0.79	1 (4)	0.46	0.05–3.97	0.48	0.77
	Cad	−	84 (94)	84 (99)					76 (93)					23 (92)				
		+	6 (6)	1 (1)	0.17	0.02–1.50	0.11	0.56	6 (7)	1.34	0.40–4.43	0.62	0.82	2 (8)	1.50	0.27–8.11	0.63	0.77
	Omp	−	92 (92)	83 (98)					75 (91)					22 (88)				
		+	8 (8)	2 (2)	0.14	0.01–1.23	0.07	0.56	7 (9)	1.22	0.40–3.72	0.71	0.82	3 (12)	1.39	0.31–6.26	0.66	0.77

Logistic regression analysis adjusted by age and sex. q-value: *p*-value adjusted after false discovery rate. Ab: antibody; OR: odds ratio; CI: confidence interval; CRC: colorectal cancer; *H. pylori: Helicobacter pylori*; +: positive; −: negative; statistically significant *p-* and q-values are indicated in bold.

**Table 3 pathogens-13-00897-t003:** IgG seropositivity to individual *H. pylori* proteins and association with development of polyp, advanced adenoma, and CRC.

Secondary Ab	Antigen		Control (n = 100) n (%)	Polyp (n = 85) n (%)	OR	95% CI	*p*-Value	q-Value	AA (n = 82) n (%)	OR	95% CI	*p*-Value	q-Value	CRC (n = 25) n (%)	OR	95% CI	*p*-Value	q-Value
**IgG**	>3 proteins	**−**	53 (53)	37 (44)					30 (37)					10 (40)				
		**+**	47 (47)	48 (56)	1.31	0.72–2.38	0.38	0.38	52 (63)	**1.88**	**1.03–3.47**	**0.04**	0.06	15 (60)	1.89	0.75–4.98	0.18	0.24
	GroEl	**−**	42 (42)	27 (32)					37 (45)					10 (40)				
		**+**	58 (58)	58 (68)	1.55	0.83–2.88	0.16	0.39	45 (55)	0.77	0.42–1.42	0.41	0.74	15 (60)	1.22	0.47–3.12	0.67	0.83
	UreA	**−**	70 (70)	55 (65)					55 (67)					17 (68)				
		**+**	30 (30)	30 (35)	1.05	0.55–2.02	0.87	0.96	27 (33)	1.12	0.59–2.15	0.71	0.86	8 (32)	1.06	0.39–2.86	0.89	0.89
	HP0231	**−**	83 (83)	69 (81)					67 (82)					17 (68)				
		**+**	17 (17)	16 (19)	1.01	0.47–2.18	0.96	0.96	15 (18)	0.97	0.44–2.12	0.94	0.94	8 (32)	1.81	0.64–5.06	0.25	0.83
	NapA	**−**	70 (70)	63 (74)					52 (63)					16 (64)				
		**+**	30 (30)	22 (26)	0.65	0.33–1.28	0.22	0.39	30 (37)	1.19	0.63–2.24	0.58	0.82	9 (36)	1.28	0.49–3.31	0.60	0.83
	HP0305	**−**	79 (79)	56 (66)					60 (73)					19 (76)				
		**+**	21 (21)	29 (34)	**2.65**	**1.31–5.36**	**0.007**	0.11	22 (27)	2	0.95–4.20	0.06	0.52	6 (24)	1.88	0.57–6.12	0.29	0.83
	HpaA	**−**	91 (91)	80 (94)					67 (81)					22 (88)				
		**+**	9 (9)	5 (6)	0.63	0.20–2.01	0.44	0.71	15 (19)	**2.46**	**1.00–6.01**	**0.048**	0.52	3 (12)	1.62	0.38–6.80	0.50	0.83
	CagA	**−**	56 (56)	34 (40)					39 (48)					12 (48)				
		**+**	44 (44)	51 (60)	**1.83**	**1.01–3.32**	**0.046**	0.18	43 (52)	1.36	0.75–2.48	0.3	0.74	13 (52)	1.49	0.59–3.74	0.39	0.83
	HyuA	**−**	77 (77)	62 (73)					58 (71)					15 (60)				
		**+**	23 (23)	23 (27)	1.1	0.55–2.19	0.77	0.96	24 (29)	1.28	0.65–2.53	0.46	0.74	10 (40)	2.02	0.77–5.27	0.15	0.83
	Catalase	**−**	81 (81)	54 (64)					57 (70)					22 (88)				
		**+**	19 (19)	31 (36)	**2.16**	**1.09–4.29**	**0.027**	0.18	25 (30)	1.47	0.71–3.03	0.29	0.74	3 (12)	0.59	0.15–2.24	0.43	0.83
	VacA	**−**	83 (83)	72 (85)					68 (83)					23 (92)				
		**+**	17 (17)	13 (15)	0.89	0.40–1.99	0.78	0.96	14 (17)	1.13	0.51–2.50	0.75	0.86	2 (8)	0.49	0.10–2.33	0.37	0.83
	HcpC	**−**	84 (84)	68 (80)					65 (79)					18 (72)				
		**+**	16 (16)	17 (20)	1.2	0.56–2.60	0.62	0.96	17 (21)	1.36	0.63–2.94	0.43	0.74	7 (28)	2.23	0.76–6.51	0.14	0.83
	Cad	**−**	89 (89)	82 (96)					72 (88)					22 (88)				
		**+**	11 (11)	3 (4)	0.25	0.06–0.94	0.04	0.39	10 (12)	1.09	0.43–2.73	0.85	0.91	3 (12)	1.16	0.29–4.68	0.82	0.88
	Omp	**−**	55 (55)	46 (54)					42 (51)					12 (48)				
		**+**	45 (45)	39 (46)	0.98	0.54–1.77	0.95	0.96	40 (49)	1.16	0.64–2.11	0.61	0.82	13 (52)	1.64	0.64–4.21	0.29	0.83

Logistic regression analysis adjusted by age and sex. q-value: *p*-value adjusted after false discovery rate. Ab: antibody; OR: odds ratio; CI: confidence interval; AA: advanced adenoma; *H. pylori: Helicobacter pylori*; +: positive; −: negative; statistically significant *p-* and q-values are indicated in bold.

## Data Availability

The original contributions presented in the study are included in the article/Appendix A, further inquiries can be directed to the corresponding author.

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
