# Peer review of "Association of Antibody Responses to Helicobacter pylori Proteins with Colorectal Adenoma and Colorectal Cancer"

_pathogens, 2024, doi:10.3390/pathogens13100897_

Round 1
Reviewer 1 Report
Comments and Suggestions for Authors
This is an investigation assessing whether antibody responses to H. pylori proteins are associated with the stages of neoplasia development, from small polyps to adenomas and advanced cancers. This article is well written and easy to understand. The methodology is reproducible.
The results are well presented, and the conclusions are consistent.
Some aspects of form need to be revised. I quote the following:
Table 1: What do “na” and “Na” mean?
Line 219: The authors state that, however, when the analysis was restricted to the NAD group, all the observed associations did not maintain statistical significance (Table S4). However, in Table S4, the a-IgG antibody responses to GroEl have a p=0.05. The authors should check that this is not a typographic mistake, otherwise the results need to be adjusted.
Line 227: The authors observed that positive correlations were found between H. pylori abundance in normal mucosa and IgA antibody responses to Catalase and VacA antigens (r = 0.48, p < 0.01; r = 0.37, p = 0.04) (Table S5). What about HyuA (r = 0.36, p = 0.05)? (Table S5).
Author Response
Point 1: This is an investigation assessing whether antibody responses to H. pylori proteins are associated with the stages of neoplasia development, from small polyps to adenomas and advanced cancers. This article is well written and easy to understand. The methodology is reproducible.
The results are well presented, and the conclusions are consistent.
Some aspects of form need to be revised. I quote the following:
Table 1: What do “na” and “Na” mean?
Response 1: In Table 1, “na” and “NA” both meant “not applicable”. We apologize and thank the reviewer for spotting this error. The abbreviation for “na” has been provided in the footnote of the table and it has been uniformly converted to “na”.
Point 2: Line 219: The authors state that, however, when the analysis was restricted to the NAD group, all the observed associations did not maintain statistical significance (Table S4). However, in Table S4, the a-IgG antibody responses to GroEl have a p=0.05. The authors should check that this is not a typographic mistake, otherwise the results need to be adjusted.
Response 2: We apologize for the oversight in missing a couple of the findings at a p=0.05 that met our significance cut off, as stated in the materials and methods section (2.5. Statistical analysis, line 186), of ≤ 0.05. Our thanks and appreciation to the reviewer for again spotting this. We have revised the text accordingly. The results section 3.1, line 217, and 219-223 now reflect the updated results: “Additionally, new statistically significant inverse and positive associations were observed with polyp development for the IgA response to HP1564 (OR = 0.09, CI: 0.009-0.98) and the IgG response to GroEL (OR = 2.23, CI: 0.97-5.12), respectively (Table S4). Although these associations were non-significant when using the full control group, they were in the same respective risk directions (Table 2).”.
The results have also been highlighted in the updated Table S4.
Point 3: Line 227: The authors observed that positive correlations were found between H. pylori abundance in normal mucosa and IgA antibody responses to Catalase and VacA antigens (r = 0.48, p < 0.01; r = 0.37, p = 0.04) (Table S5). What about HyuA (r = 0.36, p = 0.05)? (Table S5).
Response 3: We apologize for the oversight and appreciate the reviewer’s attention to detail. The Results section (3.2, lines 230-231) has been updated to include the IgA antibody response to HyuA. Additionally, the changes have been incorporated into the updated Table S5.
The corrected text now reads: “Positive correlations were found between H. pylori abundance in the normal mucosa with both the IgA and IgG antibody responses to Catalase and VacA antigens (r = 0.48, p < 0.01; r = 0.37, p = 0.04; r = 0.51, p < 0.01; r = 0.37, p = 0.04, respectively), as well as the IgA-only antibody response to HyuA (r = 0.36, p = 0.05) (Table S5).”
Additional changes:
We have included the definition of seropositivity in the abstract, line 19 and 20. This would better help the reader understand the criteria used to classify individuals as seropositive and enhance the clarity of the study's findings.
Reviewer 2 Report
Comments and Suggestions for Authors
After reading the content of the original article entitled “Association of antibody responses to Helicobacter pylori proteins with colorectal adenoma and colorectal cancer”, I believe that the main aim of the study is important and the article is written in a correct manner. Although the research is not very extensive, it is properly conducted and presents an accessible interpretation of the data obtained. To further increase the quality of the manuscript, I would like to present a short list of amendments:
- Lines 54-58: a very long sentence; please split this into two
- Table 1: 1) please add “%” close to the values (where needed) or change this in a similar way as presented in the Table 2 and 3 – n (%); otherwise its quite confusing to read,
2) “na” and “Na” – please choose one to make it uniform
- Table 2 and 3: please change “negative” and “positive” in all pleases to “-“ and “+” as it will be easier to read the results
- Line 243: “Enterotoxigenic” should be written without italics
- Lines 292 and 313: blank spaces detected, please remove
Author Response
We very much appreciate and thank you for your time and insight in reviewing this manuscript. The responses to your kind queries and suggestions are provided below, with your comments italicized and our responses marked in red. The corresponding line number in the revised manuscript is also provided with the relevant text also marked in red in the revised manuscript.
Point 1: After reading the content of the original article entitled “Association of antibody responses to Helicobacter pylori proteins with colorectal adenoma and colorectal cancer”, I believe that the main aim of the study is important, and the article is written in a correct manner. Although the research is not very extensive, it is properly conducted and presents an accessible interpretation of the data obtained. To further increase the quality of the manuscript, I would like to present a short list of amendments:
Lines 54-58: a very long sentence; please split this into two
Response 1: We agree that this sentence was over-long and thank the reviewer for suggesting this. have revised the sentence as suggested by the reviewer. The updated lines 55-60, now read: “Although H. pylori infection is mostly observed in the stomach, increasing epidemiological evidence also suggests a link with various extragastric diseases, including CRC [6]. This latter association was demonstrated experimentally by an in vivo study showing that H. pylori infection accelerated tumour growth in adenomatous polyposis coli mouse models (Apc +/- and Apc +/1638N) [7]” .
Point 2: Table 1, please add “%” close to the values (where needed) or change this in a similar way as presented in the Table 2 and 3 – n (%); otherwise its quite confusing to read,
Response 2: Our thanks to the reviewer for adding clarity to these entries. Table 1 has been modified to clearly show the percentage values.
Point 3: Table 1, “na” and “Na” – please choose one to make it uniform
Response 3: Table 1 has now been modified to uniformly display “na” in the appropriate columns.
Point 4: Table 2 and 3: please change “negative” and “positive” in all pleases to “- “and “+” as it will be easier to read the results
Response 4: We have updated Tables 2 and 3 to replace "Positive" and "Negative" with "+" and "-", respectively, as per the reviewer's suggestion. These changes have also been applied to Supplementary Tables 2-4. Additionally, the footnotes in Tables 2 and 3, as well as Supplementary Tables 2-4, now include abbreviations explaining that "+" represents "Positive" and "-" represents "Negative."
Point 5: Line 243: “Enterotoxigenic” should be written without italics
Response 5: Indeed, the reviewer is correct, and we thank the reviewer for spotting our error. The italics font of “enterotoxigenic” has been removed, the changes are now reflected in line 245.
Point 6: Lines 292 and 313: blank spaces detected, please remove
Response 6: The blank space between “who” and “completed” in line 294 has been removed, and in line 315, space at the start of the sentence, before “However” has also been removed.
Additional changes:
We have included the definition of seropositivity in the abstract, line 19 and 20. This would better help the reader understand the criteria used to classify individuals as seropositive and enhance the clarity of the study's findings.
Reviewer 3 Report
Comments and Suggestions for Authors
This is a well designed and conducted study of the association between H. pylori serum antigens and colorectal neoplasia. The existing literature is well described and the inconsistencies and contradictions in the existing literature highlighted. The results found in this study also show some inconsitency, both compared to the existing literature and within the datasets of this study, but this is acknowledged clearly and sensible hypotheses are drawn regarding how this might come about without overstepping the available evidence.
Comments on the Quality of English LanguageThe article is well written, concise and easily read. There are very few errors of english language. The sentence ending line 58 is convoluted and needs to be divided into two separate statements. Table 1 needs to be tidied for clarity with the use of "na" or "NA" but not the mixture seen. Otherwise I didn't detect any significant language problems.
Author Response
We very much appreciate and thank you for your time and insight in reviewing this manuscript. The responses to your kind queries and suggestions are provided below, with your comments italicized and our responses marked in red. The corresponding line number in the revised manuscript is also provided with the relevant text also marked in red in the revised manuscript.
Point 1: This is a well designed and conducted study of the association between H. pylori serum antigens and colorectal neoplasia. The existing literature is well described and the inconsistencies and contradictions in the existing literature highlighted. The results found in this study also show some inconsitency, both compared to the existing literature and within the datasets of this study, but this is acknowledged clearly and sensible hypotheses are drawn regarding how this might come about without overstepping the available evidence.
Comments on the Quality of English Language
The article is well written, concise and easily read. There are very few errors of english language. The sentence ending line 58 is convoluted and needs to be divided into two separate statements.
Response 1: We agree with the reviewer and apologize for this convoluted sentence. The sentence has been divided into two separate statements. The updated lines 55-60, now read: “Although H. pylori infection is mostly observed in the stomach, increasing epidemiological evidence also suggests a link with various extragastric diseases, including CRC [6]. This latter association was demonstrated experimentally by an in vivo study showing that H. pylori infection accelerated tumour growth in adenomatous polyposis coli mouse models (Apc +/- and Apc +/1638N) [7]” .
Point 2: Table 1 needs to be tidied for clarity with the use of "na" or "NA" but not the mixture seen. Otherwise, I didn't detect any significant language problems.
Response 2: Table 1 has now been modified to uniformly display “na” in the appropriate columns.
Additional changes:
We have included the definition of seropositivity in the abstract, line 19 and 20. This would better help the reader understand the criteria used to classify individuals as seropositive and enhance the clarity of the study's findings.